# Interactive Video Games as a Method to Increase Physical Activity Levels in Children Treated for Leukemia

**DOI:** 10.3390/healthcare10040692

**Published:** 2022-04-06

**Authors:** Aleksandra Kowaluk, Marek Woźniewski

**Affiliations:** Department of Physiotherapy in Surgical Medicine and Oncology, Faculty of Physiotherapy, Wroclaw University of Health and Sport Sciences, 51-612 Wroclaw, Poland; marek.wozniewski@awf.wroc.pl

**Keywords:** childhood cancer, children, cardiorespiratory fitness, physical activity, energy expenditure, interactive video games, Children’s Effort Rating Table

## Abstract

Despite the beneficial effect of exercise, children treated for cancer do not engage in sufficient physical activity. It is necessary to search for attractive forms of physical activity, including interactive video games (IVGs). The aim of this study was to verify the effectiveness of the rehabilitation model developed by the authors based on the use of IVGs in children undergoing leukemia treatment. The study included a group of 21 children aged 7–13 years (12 boys, 9 girls) undergoing treatment for acute lymphoblastic leukemia (ALL) (*n* = 13) and acute myeloid leukemia (AML) (*n* = 8). The children were randomly assigned to an intervention group and a control group. To assess the level of cardiorespiratory fitness (CRF), each child participated in a Cardiopulmonary Exercise Test. Daily physical activity was assessed using the HBSC questionnaire. The study also used the Children’s Effort Rating Table Scale (CERT) to assess the intensity of physical effort. The children in the intervention group participated in 12 sessions of. The study participants managed to complete all stages of a progressive training program, which confirmed the feasibility of such physical effort by patients with cancer. Pediatric patients reported that the IVG training required a light to moderate physical effort despite high values of energy expenditure (EE).

## 1. Introduction

The use of modern treatment methods in pediatric oncology has resulted in an increase in the survival rates of pediatric patients. Even 85% of children treated for cancer are permanently cured [1,2]. Therefore, it is necessary to identify factors that directly affect the loss of overall psychophysical fitness in pediatric patients [3,4]. Oncological treatment may cause a significant deterioration of physical and mental health. Children most often complain of fatigue, pain, anxiety, stress, social isolation [5,6], or deterioration of cardiorespiratory fitness (CRF) and physical fitness [4,7]. Such deficits persist even after treatment completion [8]. Many studies have confirmed that prolonged hospitalization and related procedures (i.e., chemotherapy, radiotherapy, surgical treatment) significantly reduce the quality of life of children and discourage them from undertaking spontaneous physical activity (PA) [9]. Studies have shown that exercise programs had a beneficial effect on the well-being and mood of pediatric patients and improved CRF and PA [10,11]. Fiuza-Luces et al. (study group, *n* = 24) proved that an intrahospital exercise program in pediatric cancer patients could be safely applied to improved muscle strength [12]. Studies have also shown that exercise interventions in this group of patients are beneficial during and after treatment [13]. Despite the beneficial effect of exercise, children treated for cancer do not engage in sufficient physical activity. Rehorst-Kleinlugtenbelt et al. (study group, *n* = 25) proved that children undergoing cancer treatment, both in hospital and in home settings, showed a reduced PA level and did not comply with the general recommendations for PA in children [14]. As a result, it is necessary to search for modern forms of physical activity attractive for children, including interactive video games (IVGs) [15].

To date, IVGs used to increase physical activity levels have been widely used in healthy children and young adults, overweight and obese children, as well as adults [16,17,18,19,20,21]. IVGs have also been applied in a group of children with developmental disorders and abnormal motor patterns. Interactive games positively influenced the improvement of the examined parameters: small and large motor skills, balance, coordination, natural forms of movement and locomotion (running, walking, jumping) [22]. IVGs have also been used as part of a rehabilitation program for children with cerebral palsy and in the case of amputation [23,24,25]. IVGs have not been commonly used in children undergoing treatment for malignant tumors [15,26].

The aim of this study was to verify the effectiveness and feasibility of the rehabilitation model developed by the authors with the use of IVGs in children undergoing leukemia treatment. In addition, the levels of cardiorespiratory fitness, physical activity, and sedentary behavior were assessed, during hospitalization and in a follow-up study.

## 2. Study Design

### 2.1. Participants and Recruitment

The selected sample included children diagnosed with acute lymphoblastic leukemia (ALL) or acute myeloid leukemia (AML) during hospitalization; the disease period did not exceed 6 months from the diagnosis. Patients were included from January 2019 until January 2020. The study participants comprised children undergoing treatment for cancer (cycles of chemotherapy in hospital settings) at the Department of Pediatric Bone Marrow Transplantation, Oncology and Hematology at the Wroclaw University Clinical Hospital, Poland. We recruited a group of 21 children aged 7–13 years (12 boys, 9 girls) undergoing treatment for ALL (*n* = 13) and AML (*n* = 8). The children were randomly assigned to the intervention and the control groups. The subjects from the intervention group participated in IVGs in the intrahospital intervention program. The children from the control group were not included in any rehabilitation program. The children from the control group reported no possession of any interactive video game kit. Not all of the recruited children completed the research program (Figure 1). The distribution of the recruited children into the intervention and the control groups allowed for checking whether the participation in the IVGs program improved their health behavior and regular physical activity level. Additionally, it was possible to assess whether the acquired health habits were permanent and whether they significantly improved the efficiency parameters of the children.

### 2.2. Intervention Group

The intervention group included 10 children (5 boys, 5 girls) undergoing treatment for ALL (*n* = 8) and AML (*n* = 2). The pediatric patients were aged 7–13 years (mean age 11.3, SD 2.0 years; mean body height 149.1, SD 13.76 cm; mean body weight 46.59, SD 16.0 kg) (Table 1).

### 2.3. Control Group

The control group included 11 children (7 boys, 4 girls) undergoing treatment for ALL (*n* = 5) and AML (*n* = 6). The patients were aged 7–13 years (mean age, 10.08, SD 1.9 years; mean body height, 140.0, SD 16.5 cm; mean body weight, 36.45, SD 9.9 kg) (Table 1).

### 2.4. Participants Characteristics

Height and weight were measured in each participant prior to the study. The subjects were enrolled by a physician and a physiotherapist. Both inclusion and exclusion criteria were defined. The inclusion criteria in the study group were as follows: diagnosed cancer (ALL or AML), 7–13 years of age, hospital treatment, duration of hospital stay >7 days, chemotherapy, lack of physical disability, unassisted arrival at the examination, written consent of a parent/legal guardian to participate in the study, height >120 cm. The exclusion criteria were as follows: platelet count <20,000/mm^3^, hemoglobin level <8 g/dL, infectious disease with fever >38 °C, intellectual disability. The children underwent the cycles of chemotherapy in the hospital settings. The mean time of treatment was 6.22 months (SD 1.64). None of the subjects presented with comorbidities.

The subjects treated for ALL were included in the International collaborative treatment protocol for children and adolescents with acute lymphoblastic leukemia (AIEOP-BFM ALL 2017 protocol) [27] (*n* = 13), whereas those treated for AML were included in the International therapeutic protocol for children with acute myeloid leukemia (AML-BFM 2012 protocol) [28] (*n* = 8). Depending on the type of ALL, the treatment was different, according to the B-ALL regimen (*n* = 8) or the T-ALL (*n* = 5) regimen. After the first stage of treatment (i.e., after induction), all children were classified into three risk groups based on the following criteria: age, leukocyte count, type of leukemia, response rate to treatment, remission, and cytogenetic results. Three children were enrolled in the standard-risk group, one child in the intermediate-risk group, and five children in the high-risk group.

## 3. Research Methods

### 3.1. Cardiorespiratory Fitness

Among the methods for the assessment of exercise tolerance which show the level of cardiorespiratory fitness, the most reliable and commonly used is the measurement of peak oxygen uptake (VO_2peak_) by means of respiratory gas analysis performed during the gradually increasing the load, known as the Cardio Pulmonary Exercise Test (CPET) [29]. According to the World Health Organization, this measurement is considered the gold standard to assess aerobic exercise capacity [30].

To assess the baseline level of cardiorespiratory fitness, each child (from intervention and control groups) participated in a CPET. The test was repeated after 14 months and not immediately after the intervention, as planned. The study was discontinued due to restrictions in the initial period of the SARS-CoV-2 pandemic. The test was initiated with a 3 min warm-up at 15 W (height of 120–150 cm) or 20 W (height > 150 cm). After the warm-up period, the test began. The load was increased at one-minute intervals by 15 or 20 W (depending on the patient’s height) according to the progressive Godfrey protocol [31,32]. The pedal frequency was at the constant level of 60–80 rotations per minute (RPM). The peak value of exercise was defined as the moment when one of the three following criteria was met: decrease in pedal frequency below 60 RPM, despite the strong verbal encouragement given by the investigator; HR_peak_ > 180 beats per minute; peak respiratory exchange ratio (RER_peak_) > 1.0. The peak oxygen uptake (VO_2peak_) was adopted as the mean value and was obtained during the last 30 s of the test [33]. Due to safety reasons, the children were not subjected to vigorous exercise. Additionally, the maximum oxygen uptake (VO_2max_) was not assessed, as opposed to the peak value of this parameter. The VO_2peak_ results were compared with predicted values for age and sex of the study participants [34]. A portable ergospirometry system (K4b2; COSMED) was used in the study. This system is used to measure metabolic parameters, i.e., pulmonary gas exchange and indirect calorimetry—volume of O_2_ uptake (VO_2_), volume of exhaled CO_2_ (VCO_2_), respiratory quotient (RQ), minute ventilation, heart rate (HR), and energy expenditure (EE). The ergospirometer allows the measurement of O_2_ and CO_2_ concentrations during the inspiratory and expiratory phases. The cycle ergometer ASPEL CRG200 was also used, which enabled to set the appropriate load at scheduled intervals. The cycle ergometer is designed to work with the CardioTEST stress test system and the AsTER cardiac rehabilitation system (Figure 2).

### 3.2. Children’s Effort Rating Table (CERT) Scale

The study also used a scale to assess the intensity of physical effort during the IVGs sessions (Figure 2). It is a scale that graphically depicts the level of physical effort experienced by the participants. The CERT scale is recommended for children due to reproducibility and a simple and understandable graphic design [35]. Moreover, other stages of the scale adequately represent the level of physical effort, and the results adequately correlate with heart rate and oxygen uptake [35,36].

### 3.3. Energy Expenditure during the IVGs Intervention

Energy expenditure during CPET and IVGs training was assessed by Keytel equation [37] used to estimate the amount of energy expenditure in children during the intervention (Figure 2). The following factors were considered: sex, weight, age, and heart rate.

### 3.4. Health Behavior Based on the School-Aged Children (HBSC 2018) Questionnaire

The level of physical activity in the intervention and control groups was assessed using the questions in the Health Behavior in School-Aged Children (HBSC 2018) questionnaire, in the section regarding health behavior (Figure 2). The questions were related to the last seven days [32]. The survey questions concerned the following: number of days per week in which the child performed physical activity for at least 60 min (Moderate to Vigorous Physical Activity—MVPA) (HBSC 1); frequency of undertaking vigorous physical activity (HBSC 2); number of hours in front of a screen per week (HBSC 4.1); number of hours in front of a screen in the weekend (HBSC 4.2); number of hours spent playing games per week (HBSC 5.1); number of hours spent playing games in the weekend (HBSC 5.2); number of hours spent using a computer, tablet, or smartphone per week (HBSC 6.1); number of hours spent using a computer, tablet, or smartphone in the weekend (HBSC 6.2).

## 4. Training with IVGs—Intervention Group

Children from the intervention group participated in 12 sessions of IVGs during hospitalization (Xbox 360 console, Microsoft), while the control group children received no intervention. Such IVGs sessions were held three times per week for four weeks (Figure 2). The intervention consisted of intervals and began with a 3 min introductory session during which each participant played any game to learn how the console worked. The sessions consisted in playing four different IVGs, each lasting 5 min. Between the games, the participant could rest for 1 min. The type of physical effort during the selected games was most similar to a continuous effort. The amount of exercise load during IVGs training was determined by the results of the baseline CPET. The intensity of the training (moderate level) was chosen considering the relationship between oxygen uptake and heart rate in children. The training was characterized by increasing intensity, which was adjusted by a gradual introduction of more advanced game levels to reach the target HR for each session. Strong verbal encouragement given by the researcher and the parents during the games was crucial to achieve the expected HR values [38]. Each game had three levels of difficulty that were gradually introduced during subsequent IVGs sessions. The difficulty level of the game depended on the pace of the game, the sensitivity of the console to movement, and the number of obstacles that occurred in the game. A gradual increase in these parameters required the player to increase the frequency of the motor response.

The IVGs kit included a motion sensor gaming console and a TV screen. The Kinect Xbox motion sensor is an input device that allows the user to interact with the console, without the need for a controller, through an interface using body gestures and voice commands.

During the game, the level of effort (intensity) was monitored using a physical activity monitor (Polar M 430). It was an additional tool that allowed the real-time monitoring of HR and assessed the effort intensity in children. The HR value was checked in real time using the Polar Flow App compatible with the PA monitor. In addition, the researcher asked each child to control the HR and report the achievement of the planned HR values. The following games were selected: Kinect Sports, Kinect Sports Season Two, and Kinect Adventures. The intervention regimen using IVGs (5 min exercise, 1 min break) resulted in interval effort. We distinguished three levels of difficulty (Table 2). First level—70% HR_peak_ (1–4 IVG session), second level—75% HR_peak_ (5–8 IVG session), third level—80% HR_peak_ (9–12 IVG session). The game types during each selected categories were: Beach volleyball, Tennis, River rush, Reflex ridge. Every game lasted 5 min, and every break between game periods lasted 1 min.

## 5. Ethics

The study was approved by the Local Bioethics Committee of Wroclaw University of Health and Sport Sciences, al. Ignacego Jana Paderewskiego 35, 51-612 Wrocław. Approval Code: 22/2018; Approval Date: 3 July 2018.

## 6. Statistical Analysis

Statistical analysis was performed using GraphPad Prism 7 software (Institute of Immunology and Experimental Therapy, Wroclaw, Poland). The normality of data distribution was assessed using the Shapiro–Wilk test. The parameters describing the group characteristics were given by descriptive statistics, such as arithmetic mean and standard deviation (SD). The ANOVA test with Brown–Forsythe correction for inequality of variance was used for the global analysis of differences between the groups, followed by Dunnett’s post-hoc tests to test the significance of the differences between the selected variables. Next, the Student’s *t*-test for independent groups with Welch correction was used to assess the statistical significance of the differences in the results between the examined groups and age- and sex-predicted values.

To demonstrate the statistical significance of differences in CPET test results before and after the intervention in both groups (intervention and control groups), we used the ANOVA test with Brown–Forsythe correction for inequality of variance for the global analysis of differences between the groups and then Dunnett’s post-hoc tests to test the significance of differences between selected variables. Depending on the data distribution, the Student’s *t*-test with Welch’s correction was used for data with a normal distribution. When the distribution was different from normal, the Dunn’s test was used.

The Student’s *t*-test with Welch’s correction was used to assess the statistical significance of differences in the results of energy expenditure between the groups of boys and girls.

To verify the statistical significance of the differences in the HBSC survey results between the intervention and the control groups, the Kruskal Wallis test was used for a global analysis of differences between the groups and then Dunn’s post-hoc tests to test the significance of differences between the selected variables.

The Wilcoxon test was used to verify the differences in the level of physical activity (before the examination vs. immediately after the intervention; after the intervention and in the follow-up study after 14 months) within the groups. The level of significance was adopted at *p* < 0.05.

## 7. Results

### 7.1. Cardiorespiratory Fitness before the IVGs Intervention—CPET Results

The mean value of VO_2peak_, which was measured during CPET, was 22.5 mL kg^−1^ min^−1^ (SD 2.6) in the intervention group of children. The mean value of this parameter was 23.3 mL kg^−1^ min^−1^ (SD 2.7) in the group of boys and 21.69 mL kg^−1^ min^−1^ (SD 2.5) the group of girls (Table 3). In the control group, the mean value of VO_2peak_ was 21.86 mL kg^−1^ min^−1^ (SD 2.44). In the control group of boys, the mean value of this parameter was 22.22 mL kg^−1^ min^−1^ (SD 2.8), and in the control group of girls it was 21.24 mL kg^−1^ min^−1^ (SD 1.9) (Table 3). The mean VO_2peak_ predicted for this age group was 45.48 mL kg^−1^ min^−1^ (SD 3.8). The predicted value of the VO_2peak_ parameter in the group of healthy boys was 46.3 (SD 4.2), while in the group of healthy girls it was 44.7 (SD 3.4) (Figure 3) [34]. The absolute difference of the measured and predicted VO_2peak_ in the group of childhood cancer patients and the predicted values in healthy children was 23.32 mL kg^−1^ min^−1^. In the groups of boys and girls, the difference was 23.3 mL kg^−1^ min^−1^ and 23.22 mL kg^−1^ min^−1^, respectively (Figure 3).

### 7.2. Assessment of the Feasibility of Training with IVGs in the Intervention Group

The assessment of heart rate in the different training phases showed that the assumed training heart rate values were achieved in the group of girls, which was particularly evident in the final stages of the consecutive IVGs training phases. The required heart rate values for each training phase (70% HR_peak_, 75% HR_peak_, and 80% HR_peak_) were achieved, and in some cases these values were even exceeded, and the children achieved higher HR values than predicted. Then children reached the assumed HR values for each phase mostly during the second or third training of the series. In the final stage of the training, all girls achieved the assumed HR values, which means that the training was feasible in the group of girls (Figure 4).

The analysis of heart rate values in different training phases showed that in the group of boys, all subjects achieved the assumed HR values, or the results were close to the predicted values. The heart rate values required for each training phase (70% HR_peak_, 75% HR_peak_ and 80% HR_peak_) were achieved, particularly in the final phases of the individual training stages. In the final phase of training, all boys achieved the assumed HR values. One participant reached a value close to 80% HR_peak_. The assumptions of the training using IVGs based on the heart rate analysis in the group of boys were met (Figure 5).

The results of CERT during the training sessions with IVGs showed that physical effort was of a light to moderate level. The intervention group participants declared slight tension or light strain (Figure 6).

### 7.3. Assessment of Energy Expenditure during Training with the Use of IVGs

The assessment of energy expenditure achieved in the group of girls during the following phases of training with the use of IVGs showed the progressive nature of the intervention. In each subsequent training phase, the girls achieved higher values of energy expenditure. The values of energy expenditure reached in the initial phases of training showed a moderate intensity of physical effort (Figure 7).

Higher energy expenditure values were found in the group of boys in the early stages of training compared to the group of girls. In the final stages of training, the boys also achieved higher energy expenditure values compared to the girls (Figure 8).

The comparison of the mean values of energy expenditure achieved by the children during the pre-intervention CPET, post-intervention CPET (after 14 months), and subsequent IVGs training sessions showed that the boys reached higher values of energy expenditure compared to the girls. The values of energy expenditure achieved by the children in the CPET after 14 months showed that the children achieved much higher values of energy expenditure compared to the values measured in the baseline examination (Table 4).

### 7.4. Cardiorespiratory Fitness after 14 Months following the IVGs Intervention—CPET Results

The results of the examination conducted 14 months after the IVGs intervention indicated an improvement in the level of cardiovascular and respiratory efficiency in the intervention group. A statistically significant improvement of the following CPET parameters peak oxygen uptake and intensity of exercise determined by the size of MET and the test duration was observed. Statistically significant differences were also observed in the following morphotic parameters: HGB (hemoglobin level), PLT (blood platelet count), RBC (red blood cell count), WBC (white blood cell count). In the control group, an increase of the level of cardiovascular and respiratory efficiency was observed but it was not statistically significant. However, a statistically significant increase was observed in the parameters: HGB, PLT, RBC, WBC and in the test duration (Table 5 and Table 6).

The results of the examination conducted 14 months after the IVGs intervention in the control group indicated a statistically significant improvement in the level of HGB, PLT, RBC, WBC and in the test duration (Table 7).

The analysis of peak oxygen consumption conducted in CPET after 14 months indicated an increase level of this parameter in both groups of examined children. The values achieved by children treated for leukemia still indicated considerable deviation from the expected values (observed in the healthy children group) (Figure 9).

### 7.5. Level of Physical Activity Immediately after the IVGs Intervention and in the Follow-Up Study

The results of our study assessing the level of physical activity in children after the IVGs intervention showed statistically significant differences. Children from the intervention group more often performed physical activity. Children included in the IVGs rehabilitation program fulfilled the recommendations to undertake physical activity at least 3 times a week. There was no statistically significant difference in the level of physical activity between the intervention and the control groups in the follow-up study 14 months after the IVG intervention (Table 8).

After the IVGs intervention, the children from the intervention group were more physically active compared to the pre-intervention period. The level of physical activity in the intervention group assessed 14 months after the end of the intervention was comparable to the level of physical activity immediately after the end of the IVGs program, and the difference was not statistically significant. Children reduced the time spent in front of the screen and the time devoted to using modern technologies (internet, computer, stationary games) (Table 9).

Children in the control group in the comparative study (at the beginning of the study and after 1 month in the re-examination) did not increase their physical activity level during the hospitalization period. After hospitalization, the children in the control group increased their daily physical activity, as it was shown by the results of the follow-up examination after 14 months (Table 10).

## 8. Discussion

The physiological assessment of the cardiovascular function, during which respiratory gases and heart rate were analyzed, showed that children treated for cancer presented with significantly reduced cardiorespiratory fitness compared to healthy peers [34], which was confirmed by the low VO_2peak_ values during CPET (Table 3 and Table 5). The mean value of VO_2peak_, which was measured during baseline CPET, was 22.5 mL kg^−1^ min^−1^ (SD 2.6) in the intervention group and 21.86 mL kg^−1^ min^−1^ (SD 2.44) in the control group. After 14 months, the absolute difference of the measured and predicted VO_2peak_ values was 20.13 mL kg^−1^ min^−1^ in the intervention group and 22.68 mL kg^−1^ min^−1^ in the control group. The VO_2peak_ values were still significantly low (Figure 3 and Figure 9).

The decreased level of baseline CPET was due to low blood count values, especially in terms of hemoglobin (Table 1). These values allow physical effort but to a limited extent. As a result, the children experienced fatigue more quickly due to insufficient oxygen delivery to muscles [39,40].

VE/VCO_2_ is another important limiting factor when undertaking a great physical effort. The assessment of the relationship between ventilation and carbon dioxide (VE/VCO_2_) is recommended and has a particular prognostic value in heart failure, which is a relatively common adverse effect of cancer treatment. Children treated for leukemia often develop cardiac disorders related to chemotherapy [41,42]. The results of the baseline CPET showed high values of this parameter in both groups. Another assessment conducted after 14 months also showed high values of the VE/VCO_2_ ratio in the group of children treated for leukemia. The assessment of the VE/VCO_2_ ratio in this group of patients may be an important prognostic factor. Additionally, special attention should be paid to the applied form of physical activity and rehabilitation, depending on the stage of cancer treatment.

The results of the baseline CPET assessment allowed for selecting individual IVGs training parameters for each participant in the intervention group. This is an innovative method for the rehabilitation of children during cancer treatment and ensures safe IVGs training programs. Individually tailored training programs are particularly beneficial [43]. Based on the peak heart rate value achieved during the CPET, training heart rate values were calculated individually for each participant. Although IVGs are not always effective in increasing the level of physical activity in healthy children [44], they have been shown to be more effective in children during cancer therapy [15,45]. Due to isolation and a high risk of infection, hospitalized children have limited opportunities to enjoy widely available forms of physical activity [46]. Hospitalized children willingly participate in IVGs activities as they provide interesting entertainment and a temporary distraction from painful and uncomfortable medical procedures [47]. Furthermore, children’s playability increases their motivation and enjoyment [48]. Our results confirmed that most children treated for leukemia achieved the heart rate values (70% HR_peak_, 75% HR_peak_, and 80% HR_peak_) required in each subsequent training phase. In some cases, the assumed values were even exceeded. This is particularly noticeable in the final stages of the individual IVG training phases, which means that the training was feasible, and the participants only needed time to adapt (Figure 4 and Figure 5).

Our assessment of the intensity of physical effort using the CERT scale showed that the participants made a light to moderate physical effort during the IVG training. This type of physical effort is particularly recommended for children treated for cancer [22]. Although the participants did not subjectively feel a high level of fatigue during the IVG training sessions, as evidenced by the CERT scores (Figure 6), they achieved significant values of energy expenditure (Table 4). Biddiss et al. and Barnett et al. also found that children achieved a mild to moderate level of physical activity during IVGs [17,49], which is in line with recommendations during treatment [50]. Additionally, studies showed that real-time controlled and monitored interventions offered significantly more favorable results compared to simulated free play [51]. Therefore, it seems reasonable to apply individually tailored and real-time controlled training sessions using IVGs for children undergoing hospitalization and cancer treatment.

The attractive form of exercise during IVGs meant that our study participants treated the training task as fun and did not want to terminate the effort too early, which was a problem during the initial CPET. Of note, the duration of the CPET in the intervention and control groups was relatively short, and the test was completed when moderate heart rate values were achieved (Table 3 and Table 5). In the follow-up study 14 months after the IVGs intervention, the test duration was prolonged, and the peak heart rate was higher (Table 5, Table 6 and Table 7). The results of the HBSC study carried out immediately after the IVGs intervention and in the follow-up studies showed no statistically significant differences in PA level. This was due to the fact that the level of PA of the examined children was comparable in these two periods (Table 8, Table 9 and Table 10). Parents who accompanied their children in IVGs training learnt that PA was possible for their child during cancer treatment. Parents were familiarized with the MVPA guidelines and probably motivated their children to practice PA regularly even during cancer treatment.

The level of children’s motivation to undertake physical exercise in the form of a test and their fear of adverse effects were important at the beginning of the study. Children undergoing cancer treatment experience significantly reduced self-esteem [52]. Any form of assessment, including CPET, is an unpleasant and even stressful experience for them. All these limitations seem to be an obstacle to the performance of physical activity by children treated for cancer. Our research showed that the children were initially afraid to undertake an increased physical effort (Figure 4 and Figure 5), but during the IVGs training period, they found that they were able to undertake PA. The results of many researchers, including our results, confirm that IVGs training is safe and feasible in this group of patients [53].

The rehabilitation of children with cancer is essential, especially during the hospitalization period. However, the selection of appropriate forms of exercise is problematic. The study results confirmed that individually tailored exercise programs using IVGs and based on the CPET results are feasible and provide significant benefits in terms of the psychophysical health of pediatric patients.

## 9. Strengths and Limitations of This Study

The study was conducted in a relatively small group of children. However, considering the intensity of the treatment process and related complications, the group was still large in terms of size, also due to the prevalence of cancer in children, which is generally low. Statistics on cancer incidence in Poland related to children show that the group of children we assessed was large. In Poland, about 1200 new cases of cancer are diagnosed annually among children. From this group, about 360 children are diagnosed with leukemia. There are about 500,000 children in the province where our center is located. The incidence rate of leukemia is 4 children per 100,000. The analysis of the data showed that there are 20 new cases of childhood leukemia per year in the Lower Silesia Province.

All children who were finally included in the study managed to complete a 4-week training program with IVGs. Cardiorespiratory fitness testing in children undergoing chemotherapy due to leukemia is difficult to perform because of the adverse treatment-related sequelae (i.e., anemia, musculoskeletal complications, and circulatory failure resulting from the cardiotoxic effect of the drugs). However, it was conducted on a group of 10 children. The research plan assumed the use of many research tools, including ergospirometry, CERT scale, HBSC questionnaire, heart rate measurements, and the estimation of energy expenditure during the IVG intervention. to the study period was performed during the SARS-CoV-2 pandemic, the researchers were deprived of the possibility of re-testing the cardiorespiratory efficiency of the children immediately after the IVG intervention. Another study conducted 14 months after the end of the IVG intervention could not be useful in assessing the direct impact of IVG training on the improvement of the cardiovascular and respiratory parameters. The results only proved that another stage of cancer treatment (maintenance therapy) had a significant impact on improving the level of cardiorespiratory efficiency.

## 10. Future Research Directions

Future investigation is needed. Studies with a larger group of children should be performed. Additionally, it would be essential to assess the extent to which improvement in terms of increased daily physical activity is a permanent result of the interventions. It would also be useful to assess whether the IVG intervention contributed to the change in habits related to undertaking regular physical activity. Future research should focus on the use of new virtual technologies with total immersion. Studies could compare the results of the interventions and assess which intervention (either IVGs or virtual reality games) is more effective to increase the physical activity levels and improve the quality of life of pediatric patients.

## 11. Conclusions

IVGs training at an intensity determined based on the baseline cardiorespiratory fitness test is safe and could become part of the rehabilitation program of children treated for leukemia. The subjects from the intervention group completed all stages of the progressive training program, which proved the feasibility of such physical effort during cancer disease treatment and even during hospitalization. Moreover, the children’s subjective assessment of the severity of the required effort during IVGs showed that IVG training required a light or moderate effort, despite reaching high energy expenditure values.

In the early stage of cancer treatment, the children from the intervention group undertook physical activity during the IVGs interventions and fulfilled the MVPA recommendations. The results of the follow-up study, conducted 14 months after the IVGs program, showed that the children continued to perform regular physical activity, and their PA level was even comparable to that during the training intervention (no statistically significant difference in PA level between post-intervention and follow-up study). After 14 months of the IVGs intervention, the children were not cured as intensively as during the first stage of cancer treatment, and their fitness parameters were better, as shown by the CPET (the test duration was prolonged, and the peak heart rate was higher). As a result, the children treated for cancer had the opportunity to be much more active during this relative recovery period. This may indicate the necessity of sustained rehabilitation programs for children for the entire duration of their cancer treatment.

## Figures and Tables

**Figure 1 healthcare-10-00692-f001:**
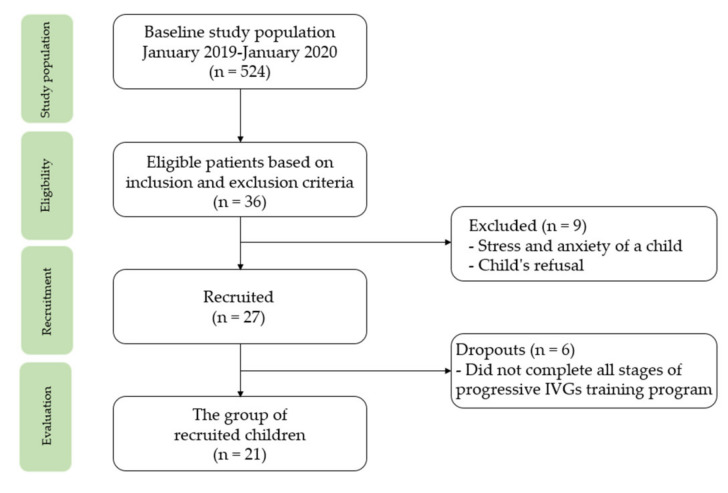
Participants recruitment.

**Figure 2 healthcare-10-00692-f002:**
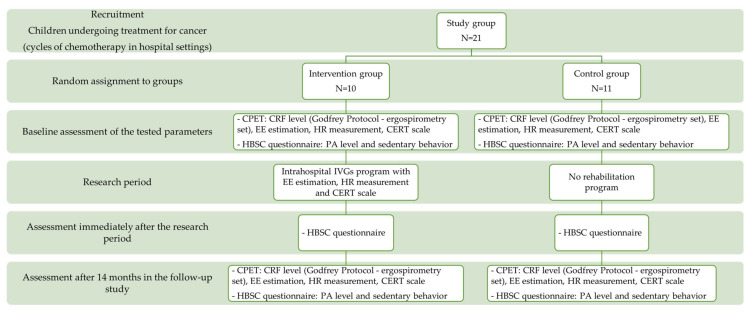
Flow chart diagram showing the scheme of the intervention carried out, with time and instruments.

**Figure 3 healthcare-10-00692-f003:**
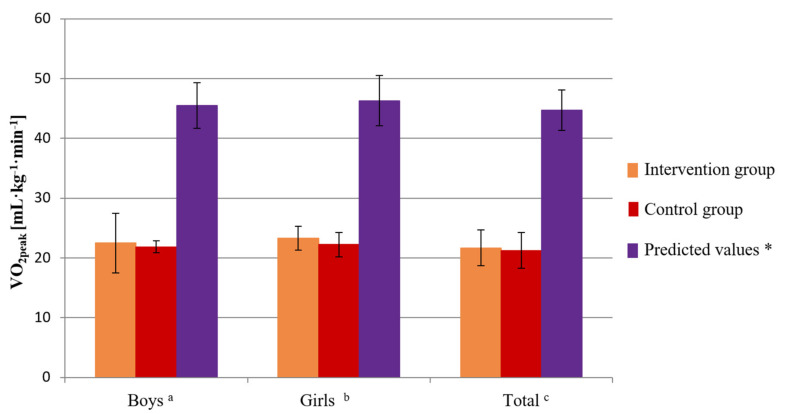
Baseline level of VO_2peak_ before the research period in the intervention and control groups compared to the predicted values for age and sex in boys, girls, and all children. Unpaired *t*-test: ^a^ boys from the intervention group vs. predicted values (*p* < 0.0001), boys from the control group vs. predicted values (*p* < 0.0001), ^b^ girls from the intervention group vs. predicted values (*p* < 0.0001), girls from the control group vs. predicted values (*p* = 0.0001), ^c^ intervention group vs. predicted values (*p* < 0.0001), control group vs. predicted values (*p* < 0.0001). * Based on age- and sex-predicted values [34].

**Figure 4 healthcare-10-00692-f004:**
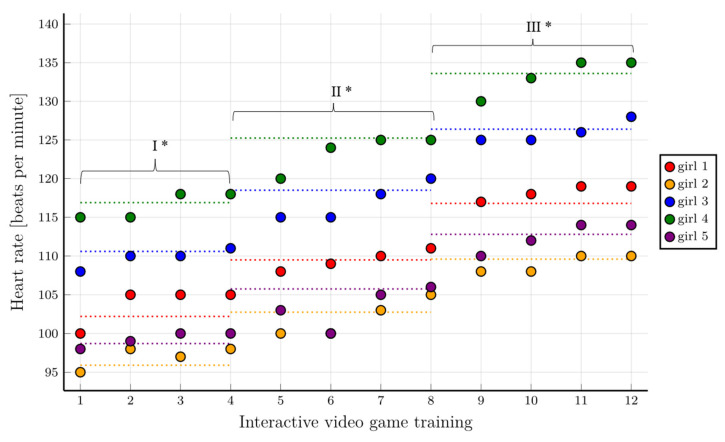
Heart rate values at different training stages in the group of girls with consideration given to individual training phases. * First level of difficulty of the game—70% HR_peak_; second level of difficulty of the game—75% HR_peak_; third level of difficulty of the game—80% HR_peak_. The horizontal dotted lines show the range of heart rate values to be achieved for a given training phase.

**Figure 5 healthcare-10-00692-f005:**
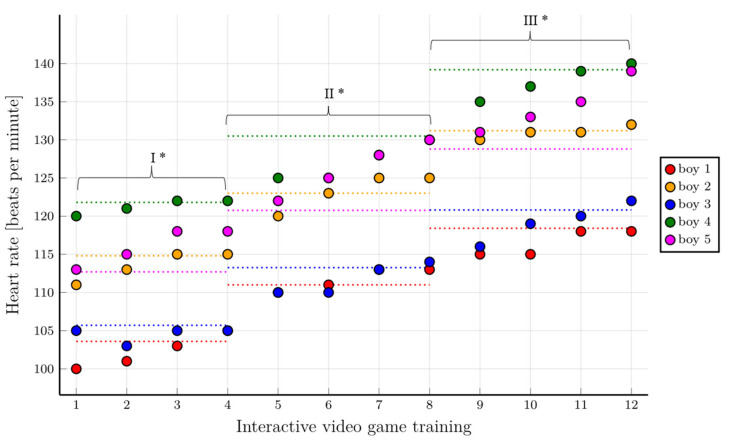
Heart rate values at different training stages in the group of boys with consideration given to individual training phases. * First level of difficulty of the game—70% HR_peak_; second level of difficulty of the game—75% HR_peak_; third level of difficulty of the game—80% HR_peak_. The horizontal dotted lines show the range of heart rate values to be achieved for a given training phase.

**Figure 6 healthcare-10-00692-f006:**
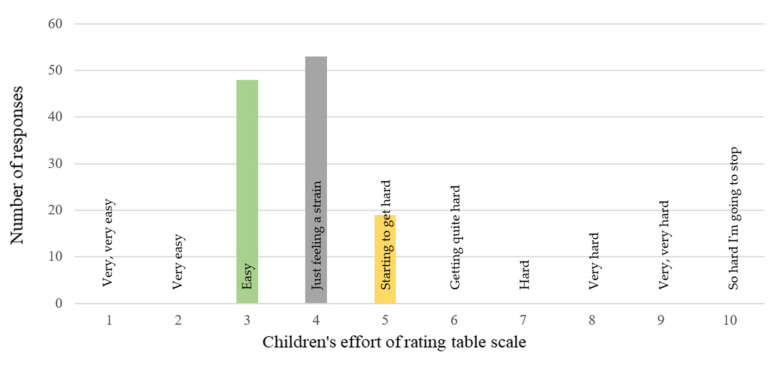
Overall assessment of the intensity of children’s effort (CERT scale) during all training sessions with IVGs.

**Figure 7 healthcare-10-00692-f007:**
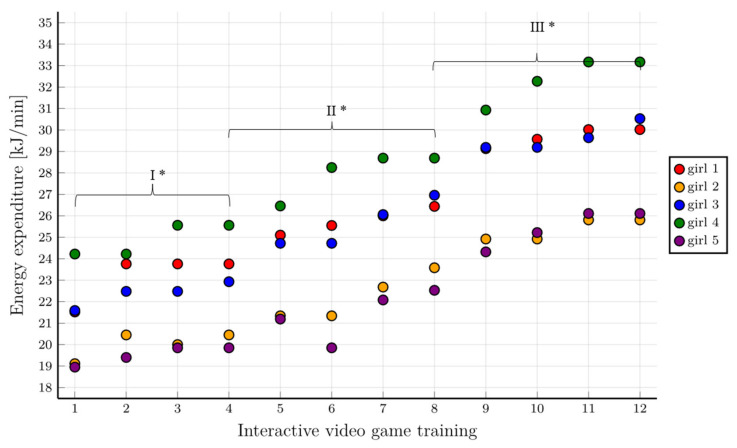
Energy expenditure in the following training stages for individual girls with consideration given to individual training phases. * First level of difficulty of the game—70% HR_peak_; second level of difficulty of the game—75% HR_peak_; third level of difficulty of the game—80% HR_peak_.

**Figure 8 healthcare-10-00692-f008:**
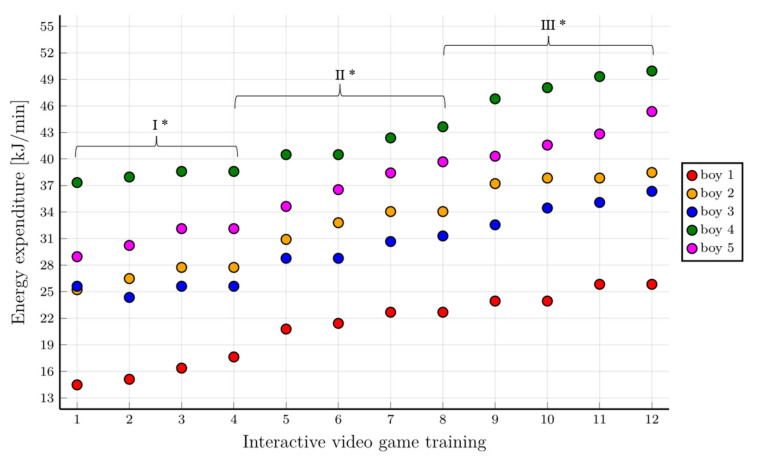
Energy expenditure in the following training stages for individual boys with consideration given to individual training phases. * First level of difficulty of the game—70% HR_peak_; second level of difficulty of the game—75% HR_peak_; third level of difficulty of the game—80% HR_peak_.

**Figure 9 healthcare-10-00692-f009:**
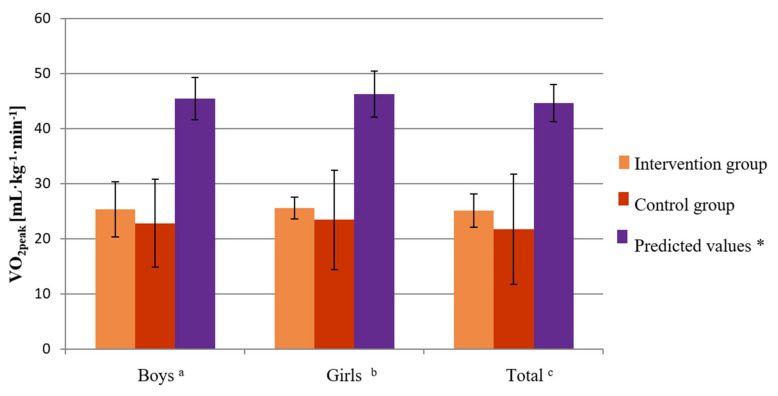
Baseline level of VO_2peak_ after the research period in the intervention and control groups compared to the predicted values for age and sex in boys, girls, and all children. Unpaired *t*-test: ^a^ values for boys in the intervention group vs. predicted values (*p* = 0.0001), values for boys in the control group vs. predicted values (*p* < 0.0001), ^b^ values for girls in the intervention group vs. predicted values (*p* < 0.0001), values for girls in the control group vs. predicted values (*p* = 0.0002), ^c^ values of the intervention group vs. predicted values (*p* < 0.0001), values of the control group vs. predicted values (*p* < 0.0001). * Based on age- and sex-predicted values [34].

**Table 1 healthcare-10-00692-t001:** Patient characteristics according to sex and group.

	Intervention Group *n* = 10			Control Group *n* = 11			*p*-Values Intervention Group vs. Control Group
Determinant Variables	Mean ± SD	Boys Mean ± SD *n* = 5	Girls Mean ± SD *n* = 5	Mean ± SD	Boys Mean ± SD *n* = 7	Girls Mean ± SD *n* = 4	
Age (years)	11.3 ± 1.9	11.4 ± 2.2	11.2 ± 2.0	10.08 ± 1.9	10.15 ± 2.0	9.9 ± 2.2	0.38
Height (cm)	149.1 ± 13.76	154 ± 15.36	144.2 ± 11.39	140 ± 16.5	141.7 ± 18.42	137 ± 14.51	0.51
Weight (kg)	46.59 ± 16.01	49.6 ± 17.8	43.5 ± 15.4	36.45 ± 9.9	36.57 ± 10.86	36.25 ± 9.43	0.24
Treatment Duration (months)	6.4 ± 1.6	6.0 ± 2.0	6.8 ± 1.3	6.3 ± 1.7	6.4 ± 2.1	6.0 ± 1.15	0.87
HGB (mg/dL)	8.35 ± 0.2	8.44 ± 0.15	8.26 ± 0.18	8.33 ± 0.17	8.24 ± 0.15	8.47 ± 0.12	0.78
PLT (G/L)	173.9 ± 57.26	162.6 ± 60.97	185.2 ± 57.8	142.3 ± 72.6	116.7 ± 76.78	187 ± 40.11	0.28
RBC (T/L)	3.143 ± 0.36	3.0 ± 0.29	3.29 ± 0.4	3.52 ± 0.29	3.46 ± 0.3	3.62 ± 0.22	0.02
WBC (G/L)	1.91 ± 0.8	2.16 ± 0.9	1.67 ± 0.57	2.27 ± 1.1	2.39 ± 1.42	2.06 ± 0.31	0.92
HR at rest	86.5 ± 3.4	86.2 ± 4.2	86.8 ± 2.9	84.9 ± 4.16	85.1 ± 4.95	84.5 ± 2.89	0.35

Note: HGB—hemoglobin level, PLT—blood platelet count, RBC—red blood cell count, WBC—white blood cell count, HR (at rest)—resting heart rate.

**Table 2 healthcare-10-00692-t002:** Interactive video game intervention—selected games.

Level of Difficulty	IVG Session Number	Game Type	Game Duration/Break Duration in Each Session
First level—70% HR_peak_	1–4	Beach volleyball	5 min/1 min
		Tennis	5 min/1 min
		River rush	5 min/1 min
		Reflex ridge	5 min/end of the game
Second level—75% HR_peak_	5–8	Beach volleyball	5 min/1 min
		Tennis	5 min/1 min
		River rush	5 min/1 min
		Reflex ridge	5 min/end of the game
Third level—80% HR_peak_	9–12	Beach volleyball	5 min/1 min
		Tennis	5 min/1 min
		River rush	5 min/1 min
		Reflex ridge	5 min/end of the game

IVGs interventions were performed by a clinical physiotherapist–researcher qualified to work with children with cancer disease. During each IVGs session, the child’s parent was also present.

**Table 3 healthcare-10-00692-t003:** Cardiorespiratory test results (CPET) before the IVGs intervention in the study and control groups.

	Intervention Group *n* = 10			Control Group *n* = 11			*p*-Values Intervention Group vs. Control Group
Outcome Variables	Mean ± SD	Boys Mean ± SD *n* = 5	Girls Mean ± SD *n* = 5	Mean ± SD	Boys Mean ± SD *n* = 7	Girls Mean ± SD *n* = 4	
VO_2peak_ (mL kg^−^^1^ min^−^^1^)	22.5 ± 2.6	23.3 ± 2.7	21.69 ± 2.5	21.86 ± 2.44	22.22 ± 2.8	21.24 ± 1.9	0.57
HR_peak_	154.7 ± 12	159.6 ± 10.45	149.8 ± 12.4	149.5 ± 10.0	143.1 ± 5.8	160.8 ± 1.7	0.30
VO_2_ (mL/min)	1068 ± 395.6	1247 ± 329.3	888. 4 ± 404.1	1228 ± 251.9	1241 ± 292.1	1205 ± 199.6	0.29
VCO_2_ (mL/min)	1069 ± 517.8	1269 ± 495.6	868.2 ± 507.3	1179 ± 403.9	1284 ± 479.1	995.8 ± 119.9	0.59
VE (L/min)	28.33 ± 9.4	31.22 ± 6.5	25.44 ± 11.7	29.47 ± 6.0	30.13 ± 6.8	28.33 ± 4.9	0.75
VE/VCO_2_	28.74 ± 6.02	26.18 ± 5.1	31.31 ± 6.3	26.8 ± 3.9	25.91 ± 4.5	28.35 ± 2.2	0.4
RQ = VCO_2exhaled_ /VO_2uptake_	0.97 ± 0.14	0.99 ± 0.16	0.94 ± 0.15	0.95 ± 0.2	1.0 ± 0.2	0.8 ± 0.06	0.93
MET	6.38 ± 0.7	6.62 ± 0.8	6.14 ± 0.7	6.2 ± 0.7	6.3 ± 0.8	6.0 ± 0.5	0.53
Test Duration (s)	481.1 ± 35.5	474.4 ± 41.1	487.8 ± 32.3	461.2 ± 32.7	469.4 ± 37.7	446.8 ± 17.2	0.95

Note: VO_2peak_—peak oxygen uptake, HR_peak_—peak heart rate, VO_2_—volume of O_2_ uptake, VCO_2_—volume of exhaled CO_2_, VE—minute ventilation, VE/VCO_2_—ventilatory equivalent of carbon dioxide, RQ—respiratory quotient, MET—metabolic equivalent of task.

**Table 4 healthcare-10-00692-t004:** Mean values of energy expenditure during baseline CPET (EE_test I_), CPET after 14 months (EE_test II_), and subsequent IVGs training sessions in the intervention group and according to sex.

	EE_test I_	EE_test II_	EE_1_	EE_2_	EE_3_	EE_4_	EE_5_	EE_6_	EE_7_	EE_8_	EE_9_	EE_10_	EE_11_	EE_12_
	Intervention group													
Mean (kJ/min)	49.83	54.59	23.7	24.44	25.21	25.42	27.44	27.97	29.37	29.95	31.93	32.7	33.57	34.16
Std. Deviation (kJ/min)	10.85	12.26	6.30	6.26	6.47	6.23	6.41	6.86	6.99	7.24	7.50	7.88	7.88	8.39
	Girls													
Mean (kJ/min)	41.92	45.22	21.08	22.06	22.33	22.51	23.76	23.94	25.1	25.64	27.7	28.23	28.95	29.13
Std. Deviation (kJ/min)	4.04	3.97	2.16	2.09	2.45	2.36	2.37	3.36	2.72	2.53	2.91	3.12	3.06	3.13
	Boys													
Mean (kJ/min)	57.74	63.96	26.32	26.82	28.09	28.34	31.11	32.0	33.64	34.27	36.16	37.17	38.18	39.19
Std. Deviation (kJ/min)	9.59	10.16	8.22	8.35	8.22	7.78	7.29	7.34	7.56	8.07	8.57	8.96	8.77	9.23
*p*-Values boys vs. girls	0.02 *	0.01 *	0.23	0.28	0.2	0.17	0.09	0.07	0.06	0.07	0.09	0.09	0.08	0.07

* Results showing the statistical significance.

**Table 5 healthcare-10-00692-t005:** Cardiorespiratory test results (CPET) after 14 months of intervention in the intervention and control groups.

	Intervention Group *n* = 10			Control Group *n* = 11			*p*-Values Intervention Group vs. Control Group
Outcome Variables	Mean ± SD	Boys Mean ± SD *n* = 5	Girls Mean ± SD *n* = 5	Mean ± SD	Boys Mean ± SD *n* = 7	Girls Mean ± SD *n* = 4	
VO_2peak_ (mL kg^−^^1^ min^−^^1^)	25.35 ± 2.3	25.59 ± 3.12	25.11 ± 1.4	22.84 ± 2.26	23.45 ± 2.24	21.78 ± 2.1	0.16
HR_peak_	163.3 ± 13.2	168.8 ± 12.3	157.8 ± 12.8	147.5 ± 16.05	139.3 ± 11.04	162 ± 13.3	0.17
VO_2_ (mL/min)	797.3 ± 250.2	830.6 ± 354.3	764 ± 111.9	1101 ± 384.1	1196 ± 441.2	936 ± 212.4	0.81
VCO_2_ (mL/min)	739.3 ± 324.6	789 ± 454.1	689.6 ± 157.2	1025 ± 295.5	1088 ± 316.8	914.8 ± 254.7	0.77
VE (L/min)	32.86 ± 7.2	32.24 ± 10.0	33.48 ± 3.8	29.05 ± 7.12	27.74 ± 5.94	31.33 ± 9.3	0.88
VE/VCO_2_	48.15 ± 14.17	45.16 ± 14.55	51.14 ± 14.8	29.16 ± 6.85	26.04 ± 2.4	34.63 ± 9.1	0.61
RQ = VCO_2exhaled_/ VO_2uptake_	0.9 ± 0.08	0.92 ± 0.1	0.89 ± 0.07	0.95 ± 0.099	0.94 ± 0.10	0.97 ± 0.1	0.90
MET	7.2 ± 0.65	7.26 ± 0.9	7.14 ± 0.4	6.58 ± 0.78	6.82 ± 0.81	6.2 ± 0.6	0.41
Test Duration (s)	591.6 ± 97.8	608.4 ± 112.9	574.8 ± 90	432.9 ± 71.7	439.14 ± 78.12	421.98 ± 84	0.62

Note: VO_2peak_—peak oxygen uptake, HR_peak_—peak heart rate, VO_2_—volume of O_2_ uptake, VCO_2_—volume of exhaled CO_2_, VE—minute ventilation, VE/VCO_2_—ventilatory equivalent of carbon dioxide, RQ—respiratory quotient, MET—metabolic equivalent of task.

**Table 6 healthcare-10-00692-t006:** Statistical significance of the differences in the CPET results between pre- and post- intervention values in the intervention group.

	VO_2peak_ ^b^ (mL kg^−^^1^ min^−^^1^)	VO_2_ ^a^ (mL/min)	VCO_2_ ^a^ (mL/min)	VE ^b^ (L/min)	VE/VCO_2_ ^a^	RQ = VCO_2exhaled_ /VO_2 uptake_ ^a^	HR_peak_ ^b^	MET ^b^	Test Duration ^b^ (s)
Mean Difference Between Pre- and Post-Intervention Results	2.86	−4.6	−8.8	4.53	7.6	−5.3	8.6	0.82	65.2
*p*-Value	0.02 *	0.72	0.5	0.24	0.56	0.68	0.14	0.02 *	0.03 *
		**HGB ^b^**		**PLT ^b^**		**RBC ^b^**		**WBC ^b^**	
Mean Difference Between Pre- and Post-Intervention Results		4.66		87,1		1.30		10.42	
*p*-Value		<0.0001 *		0.0013 *		<0.0001 *		<0.0001 *	

Note: ^a^ *p*-value of Dunn’s test, ^b^
*p*-value of the Student’s *t*-test with Welch correction, * results showing the statistical significance.

**Table 7 healthcare-10-00692-t007:** Statistical significance of the differences in the CPET results between initial examination and follow-up after 14 months in the control group.

	VO_2peak_ ^b^ (mL kg^−^^1^ min^−^^1^)	VO_2_ ^b^ (mL/min)	VCO_2_ ^b^ (mL/min)	VE ^b^ (L/min)	VE/VCO_2_ ^a^	RQ = VCO_2exhaled_/VO_2uptake_ ^a^	HR_peak_ ^b^	MET ^b^	Test Duration ^a^ (s)
Mean Difference Between Pre- and Post-Intervention Results	0.98	−126.9	−153.7	0.43	4.41	3.09	−2	0.40	124.4
*p*-Value	0.34	0.37	0.32	0.88	0.91	0.94	0.73	0.21	0.005 *
		**HGB ^b^**		**PLT ^b^**		**RBC ^b^**		**WBC ^a^**	
Mean Difference Between Pre- and Post-Intervention Results		4.34		161		0.68		80.09	
*p*-Value		<0.0001 *		<0.0001 *		0.0012 *		0.049 *	

Note: ^a^ *p*-value of Dunn’s test, ^b^
*p*-value of the Student’s *t*-test with Welch correction, * results showing the statistical significance.

**Table 8 healthcare-10-00692-t008:** Statistical significance of the differences in immediate HBSC post-intervention results and in follow-up study results conducted 14 months after the research period between the intervention and the control groups.

Kruskal-Wallis Test	Physical Activity Level Immediately after the IVGs Intervention in Intervention Group vs. Control Group							
	HBSC 1 *	HBSC 2 *	HBSC 4.1 *	HBSC 4.2 *	HBSC 5.1 *	HBSC 5.2 *	HBSC 6.1 *	HBSC 6.2 *
Difference Between Ranks	281	361.5	123	87.53	127	159.9	83.27	146.3
*p*-Value of Dunn’s Test	0.0048 **	0.0003 **	0.22	0.38	0.20	0.11	0.40	0.14
	**Physical Activity Level in Follow-Up Examination in Intervention Group vs. Control Group**							
Difference Between Ranks	194.4	130	−63.49	−142.7	1.50	−72.93	65.4	−10.97
*p*-Value of Dunn’s Test	0.12	0.28	0.6	0.24	0.99	0.54	0.59	0.93

Note: * The responses and questions were included in a previous paper [32]. The third question was not considered in the current study since the new HBSC 2018 questionnaire does not include question 3. ** Results showing statistical significance.

**Table 9 healthcare-10-00692-t009:** Statistical significance of the differences between immediate HBSC post- intervention results and results after 14 months in the intervention group.

Wilcoxon’s Test	Physical Activity Level before IVGs Intervention vs. Immediately after the IVGs Intervention							
	HBSC 1 *	HBSC 2 *	HBSC 4.1 *	HBSC 4.2 *	HBSC 5.1 *	HBSC 5.2 *	HBSC 6.1 *	HBSC 6.2 *
Median of Differences	3	4	1.5	1	1	1.5	0.5	1.5
*p*-Value Two Side Test	0.002 **	0.002 **	0.0039 **	0.0469 **	0.0156 **	0.0078 **	0.0625	0.002 **
	**Physical Activity Level Immediately after the IVGs Intervention vs. in Follow-Up Study after 14 Months**							
Median of Differences	0	0	2	3	2	2	1	1
*p*-Value Two Side Test	>0.9999	>0.9999	0.002 **	0.002 **	0.002 **	0.0039 **	0.0313 **	0.0156 **

Note: * The responses were included in the previous Table 7, and the questions in a previous paper [32]. The third question was not considered in the current study since the new HBSC 2018 questionnaire does not include question 3. ** Results showing statistical significance.

**Table 10 healthcare-10-00692-t010:** Statistical significance of the differences between immediate HBSC post- intervention results and results after 14 months in the control group.

Wilcoxon’s Test	Physical Activity Level before Examination vs. after 1 Month in the Re-Examination Study							
	HBSC 1 *	HBSC 2 *	HBSC 4.1 *	HBSC 4.2 *	HBSC 5.1 *	HBSC 5.2 *	HBSC 6.1 *	HBSC 6.2
Median of Differences	0	0	0	0	0	0	0	0
*p*-Value Two Side Test	>0.9999	>0.9999	>0.9999	>0.9999	>0.9999	0.5	>0.9999	>0.9999
	**Physical Activity Level Immediately after the IVGs Intervention vs. in Follow-Up Study after 14 Months**							
Median of Differences	1	3	3	3	4	3	2	2
*p*-Value Two Side Test	0.0039 **	0.001 **	0.001 **	0.001 **	0.001 **	0.001 **	0.002 **	0.001 **

Note: * The responses were included in the previous Table 7, and the questions in a previous paper [32]. The third question was not considered in the current study since the new HBSC 2018 questionnaire does not include question 3. ** Results showing statistical significance.

## Data Availability

The data are available upon reasonable request from the corresponding author.

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
