# Peer review of "Interactive Video Games as a Method to Increase Physical Activity Levels in Children Treated for Leukemia"

_healthcare, 2022, doi:10.3390/healthcare10040692_

Round 1
Reviewer 1 Report
Dear authors,
your paper is interesting and well written.
I think that 2 points need to be improved:
- it is not clear why you recruited 2 different groups (study and control). Please, clarify this point.
- you properly discuss the available literature. I think you should comment on any paper about this topic (if available) focusing your attention and number of patients and study design
Author Response
Thank you very much for reviewing our article. All the comments provided by you were very helpful and added value to make the text better. The answers and our comments you will find below:
- In the subsection "2. Materials and Methods; 2.1 Participants and recruitment”, we added information why we recruited two groups: study and control: “We recruited a group of 21 children aged 7-13 years (12 boys, 9 girls) undergoing treatment for ALL (n=13) and AML (n=8). The children were randomly assigned to the study and control groups. The subjects from the study group participated in IVGs in the intrahospital intervention program. The children from the control group were not included in any rehabilitation program.” Additionally, the difference in study intervention is provided on the Figure 2.
- In the “Introduction” section, we added a detailed description of previous research important for our study purpose. We did not add such a description to every cited article because we did not think it is necessary. Moreover, it would significantly increase the length of our article. In the "Introduction" section, we added a short description focused on the use of new interactive technologies in rehabilitation. We believe that such a presentation of the problem is sufficient.
Reviewer 2 Report
This study proposes a study about how video games that can be tuned to the needs of children suffering leukemia increase their physical activity level. Also, it provides validation for various example scenarios that authors developed for testing in an experimental setting with two groups. The method and validation are provided in a fairly good level of detail. However, the study will benefit from the following suggestions as there are some issues that need to be addressed:
Serious gaming literature for rehabilitation (active videogames, exergames): There is a good portion of literature that exists on the use of serious games for rehabilitation. Authors need to bring that literature into the paper and build a case for why the new method based on videogames is novel and extends beyond what has already been done in the field of rehabilitation games that is quite saturated. They need to clearly articulate literature on rehabilitation games and then progress their way into the active videogames for this purpose.
For example, several studies in the area of exergaming can be helpful for the discussion:
García-Bravo, S., Cuesta-Gómez, A., Campuzano-Ruiz, R., López-Navas, M. J., Domínguez-Paniagua, J., Araújo-Narváez, A., ... & Cano-de-la-Cuerda, R. (2021). Virtual reality and video games in cardiac rehabilitation programs. A systematic review. Disability and Rehabilitation, 43(4), 448-457.
Sousa, C. V., Hwang, J., Cabrera-Perez, R., Fernandez, A., Misawa, A., Newhook, K., & Lu, A. S. (2021). Active video games in fully immersive virtual reality elicit moderate-to-vigorous physical activity and improve cognitive performance in sedentary college students. Journal of sport and health science.
González-González, C. S., Toledo-Delgado, P. A., Muñoz-Cruz, V., & Torres-Carrion, P. V. (2019). Serious games for rehabilitation: Gestural interaction in personalized gamified exercises through a recommender system. Journal of biomedical informatics, 97, 103266.
Chiuchisan, I., Geman, O., & Postolache, O. (2018, October). Future Trends in Exergaming using MS Kinect for Medical Rehabilitation. In 2018 International Conference and Exposition on Electrical And Power Engineering (EPE) (pp. 0683-0687). IEEE.
Nin, V., Goldin, A. P., & Carboni, A. (2019). Mate Marote: Video games to stimulate the development of cognitive processes. IEEE Revista Iberoamericana de Tecnologías del Aprendizaje, 14(1), 22-31.
González, C. S. G., del Río, N. G., & Adelantado, V. N. (2018). Exploring the benefits of using gamification and videogames for physical exercise: A review of state of art. IJIMAI, 5(2), 46-52.
Hsiao, L. C., W. F. Huang, T. Y. Hsu, S. Y. Lin, H. L. Chen, and T. N. Wang. "Development of a Kinect-based bilateral rehabilitation game for children with cerebral palsy." Journal of the Neurological Sciences 381 (2017): 191-192.
The description of the method is a bit haphazard. I suggest the authors build a flow chart to report the scheme of the intervention carried out indicating periods and instruments applied and to which group. Besides, explain how the sample has been selected.
Comment on playability- Please explain how the playability/ player experience influences the adherence to physical activity of children.
Author Response
Thank you very much for reviewing our article. All the comments provided by you were very helpful and added value to make the text better. The answers and our comments you will find below:
- In the “Introduction” section, we added a description of the available literature on the use of new IVG technologies in rehabilitation and their beneficial impact on many health parameters: “To date, IVGs used to increase physical activity levels have been widely used in healthy children and young adults, in overweight and obese children as well as adults [16-21]. IVGs have also been applied in a group of children with developmental disorders and abnormal motor patterns. Interactive games positively influenced the improvement of the examined parameters: small and large motor skills, balance, coordination, natural forms of movement and locomotion (running, walking, jumping) [22]. IVGs have also been used as part of a rehabilitation program for children with cerebral palsy and in the case of amputation [23-25]. IVGs have not been commonly used in a group of children undergoing treatment for malignant tumors [15, 26]”.
This confirms that the IVG method was used in a group of children with cancer, but we performed baseline CPET assessment before IVGs intervention. Based on the results of the CPET test, we selected a rehabilitation program for each participant. This was our innovation. We included it in the “Discussion” section: “The results of the baseline CPET assessment allowed for selecting the individual IVGs training parameters for each participant in the study group. It was an innovative method of rehabilitation of children during cancer treatment and ensured safe IVGs training programs. Individually tailored training programs are particularly beneficial [43]. Based on the peak heart rate value achieved during the CPET, training heart rate values were calculated individually for each participant.”
- Thank you for providing such a large number of references proposed for citation. We took advantage of this and present the possibility of using new technologies in rehabilitation: “To date, IVGs used to increase physical activity levels have been widely used in healthy children and young adults, in overweight and obese children as well as adults [16-21]. IVGs have also been applied in a group of children with developmental disorders and abnormal motor patterns. Interactive games positively influenced the improvement of the examined parameters: small and large motor skills, balance, coordination, natural forms of movement and locomotion (running, walking, jumping) [22]. IVGs have also been used as part of a rehabilitation program for children with cerebral palsy and in the case of amputation [23-25]. IVGs have not been commonly used in a group of children undergoing treatment for malignant tumors [15, 26].”
Additionally, we used this references in the description of the term "playability" in the "Discussion" section: “Hospitalized children willingly participate in IVGs activities as they provide interesting entertainment and a temporary distraction from painful and uncomfortable medical procedures [47]. Furthermore, children's playability increases their motivation and enjoyment [48].”
- We added flowcharts (Figure 1, 2) to make the description of the methods more understandable, and the different research periods more visible.
- We added a flowchart (Figure 1.) about the recruitment of the study participants to explain how the sample was selected.
Reviewer 3 Report
Thank you for the opportunity to read an interesting article on an important issue in the field of public health. Text fits to the aim and scope of the journal.
I would like to mention some issue, which should help to improve the quality of the article.
Main issues:
- The main concern relates to the size of the study group. Please confirm using eg. GPower program, that the number of studied patients is sufficient. This is crucial information.
- I have impression that you put all collected information in one article without reflection how and why do you want to use them... In my opinion you haven't prove that the results from Cardiopulmonary Exercise Test are informative in any case regarding your intervention. Test done more than one year after intervention is affected by so many determinants, and for sure not by intervention. So, what is the reason to show them? What do you want to prove or what kind of hypothesis do you want to verify using it?
- It brings me to another issue - aim of the study, it could be extended of more precise questions/ hypothesis concerning variables which you used.
- The information about approval of the bioethics committee should be into the main text, please complete it.
- I'm not familiar with chemotherapy standards, but I would suggest giving the literature references or more information about cut point for each risk group. It is very important information for possible replication of the research (p.7 line86-94). Another question is how you used this information, you don’t refer to this aspect in any moment in the text.
- I noticed an inconsistency in the description of "Material and Methods", 2.1 study group, 2.2 control group and into this point again the description of the whole group under the table, as if it was still a control group. Please correct it.
- What was the percentage of people who did not complete the study, if any?
- There is missing some information about study protocol: where were realized training sessions eg. home or hospital? How did you control in real time the HR? Did you have connection with polar sensor on tablet or on other devises or did you ask children to look at PA monitor? How were the sessions organized, who was involved in it? Do you have information about having such equipment by children from intervention and control group?
- The Figures 2 and 3 are hardly legible; the description shows the heart rate ranges, but there is no information about how they are marked on the chart (the same in Fig. 6 and 7); what the individual colours mean; you mentioned that there are records of 12 sessions, why there is no record of the last one.
- The Fig.4 is also not informative enough; what different colour means? Maybe you could show differentiation between boys and girls?
- As I understand correctly there are repeated information in Table 4 from Fig. 3 and 4.
- I would suggest using standard format for table 6 and 7 with the same arrangement, which could help with comparison of two groups; you can put information about statistical test used for analysis with e.g. superscript letters.
- Discussion part refers to less than 10 publications; no one related to main aim of the article. Is there so little literature in this aspect? Also, discussion is written as one paragraph - it is hard to read.
- The second paragraph of the conclusions in my opinion is not prove in the study, especially that IVGs intervention changes the lifestyle of the children – it is eg connected with my last questions from point 8...
- Most of the references is quite old, especially then writing about using new technology.
- There are strange supplementary files with this text... I don't understand why they are there...
Small Issue:
p.7 line 86 "AIEOP-BFM ALL 2017" and line 87 "AML-BFM 2012" - should be explain.
Tabeles: there are prepared very carelessly, eg. Tab. 1 there are not needed underline inside it, the lines are in different thickness, Tab 2 - different lines; in my opinion "-" are not necessary before name of games...
Abbreviations - there is a big mess and lack of consistency in inserting and using abbreviations in the text, eg. on p.7 line 77, line 86 there is an abbreviation ALL and then again, the full form on line 88 and abbreviation in brackets ... It should be checked according to all abbreviations used in text and corrected.
Point 3. Research Methods - the name is not the best in my opinion, because methods should be described in point 2. Material and Methods; maybe: study design, and as a part of point 2.
3.1 and the rest is the way how you collected the data, so it is Material and Methods part...
p.7 line 98 for Cardiopulmonary Exercise Test (CPET) - references is necessary.
p.7 line 149 "The amount of exercise load during IVG training was determined by 149 the results of the exercise test" - what kind and when did the exercise? Please clarify.
Under table 3 on p.8 there is *... maybe better change it for "Note: ..."
The results in Fig.1 and Fig.7 refers to results from before intervention. Please clarify.
I think that Tab. 8 would be easier to read in different (more classical) shape or as a text. It is mentioned that **results shows the correlation, I'm confused - in title is differentiation ...?
Author Response
Thank you very much for reviewing our article. All the comments provided by you were very helpful and added value to make the text better. The answers and our comments you will find below:
Main issues:
- We are aware that our tested sample was not large, but taking into account the fact that we examined children with a specific type of disease, at the appropriate age and at the initial stage of cancer treatment, our sample, given following criteria was a large sample. We explained it in the section of “Strengths and limitations of this study”. We presented the statistics and the incidence rate for the treatment of leukemia in our region: “The study was conducted in a relatively small group of children. However, considering the intensity of the treatment process and related complications, the group was still large in terms of size, also due to the prevalence of cancer in children, which is generally low. Statistics on cancer incidence in Poland related to children show that the group of children we assessed was large. In Poland about 1200 new cases of cancer are diagnosed annually among children. From this group about 360 children are diagnosed with leukemia. There are about 500,000 children in the province (voivodeship Lower Silesia) where our center is located. The incidence rate of leukemia is 4 children per 100,000. The analysis of the data shows that there are 20 new cases of childhood leukemia per year in the Lower Silesia Province.”
In addition, we added a flow chart that shows how many patients we qualified for the study and how many children finally took part in examination – Figure 1.
- We included in the “Discussion” section why we re-examined our study participants after 14 months in the follow-up study. We analyzed whether the participation in IVGs intervention resulted in a permanent change in health behaviors: “In follow-up study 14 months after the IVGs intervention, the test duration was prolonged and the peak heart rate was higher (Tables 5, 6, 7). The results of the HBSC study carried out immediately after the IVGs intervention and in the follow-up studies showed no statistically significant differences in PA level. This is due to the fact that the level of PA of the examined children was comparable in these two periods (Table 8, 9, 10).”
Additionally, we included the information about the presence of parents was significant in the results: " Parents who accompanied their children in IVGs training learnt that PA was possible during childhood cancer treatment. Parents were familiarized with the MVPA guide-lines and they probably motivated their children to practice PA regularly even during cancer treatment.”
- We extended the aim of the study. Now, the aim includes all the areas we examined: “The aim of the study was to verify the effectiveness and feasibility of the rehabilitation model developed by the authors with the use of IVGs in children undergoing leukemia treatment. In addition, the level of cardiorespiratory fitness, physical activity level and sedentary behavior was assessed, during hospitalization and in follow-up study.”
- We added the information about the consent of the bioethics committee in the main text.
- We explained an abbreviation of used treatment protocols. We also provided literature concerning the information about cut-points for each leukemia risk groups. Information about the number of children belonging to a given risk group may be used as an element of repeatability of the research project (how many people were treated with a given protocol). Depending on the risk group, the treatment process is different. It is an important information because it is also associated with the occurrence of specific psychophysical disorders.
- We corrected the names of the chapters concerning the information about study participants and methods. We changed the name of the subsection to the: 2. Study design; 2.1 Participants and Recruitment; 2.1 Study group; 2.2. Control Group; 2.3 Participants Characteristics. The research methods are only explained in the subsection “Research Methods”.
- We added the information about the completion of the research period by participants in the main text and on the figure: “Not all of the recruited children completed the research program (Figure 1).” The flowchart further facilitates the understanding of the recruiting process.
- In the section “Study design; Participants and recruitment, we added the information about where the IVGs intervention took place: “The subjects from the study group participated in IVGs in the intrahospital intervention program.” Moreover, Figure 2 there is also contains such information.
- We added the information about how the HR was controlled in real time. Subsection “Study group intervention; Training with IVGs”: “During the game, the level of effort (intensity) was monitored using a physical activity monitor (Polar M 430). It was an additional tool that allowed the real-time monitoring of HR and assessed the effort intensity in children. The HR value was checked in real time using the Polar Flow App compatible with the PA monitor. In addition, the researcher asked each child to control HR and report the achievement of planned HR values.”
- In subsection “4. Study group intervention; Training with IVGs” we added the information about who was involved in IVGs training sessions: “IVGs interventions were performed by a clinical physiotherapist-researcher qualified to work with children with cancer disease. During each IVGs session the child's parent was also present.”
- In subsection “2. Study Design; 2.1 Participants and Recruitment” we added information about the possession of IVG equipment by children from the control group: “The children from the control group were not included in any rehabilitation program. The children from the control group reported no possession of any interactive video game kit.”
- We corrected the Figures. 4, 5 and Figures 7, 8. These figures are now renumbered. Now the assumed training heart rate values are more visible. Figure descriptions are more detailed, and each of the participants is marked with different colors, consistent in both HR and EE charts.
- We have corrected the figure with the current number 6. Now the Figure 6. contains a description of the individual grades of the CERT scale and the feelings about the severity of the effort given to the children.
- Table 4. contains the mean energy expenditure values achieved by the children in each IVGs training sessions. We corrected the name of the table: “Mean values of energy expenditure during the baseline CPET (EEtest I), CPET after 14 months (EEtest II) and subsequent IVGs training sessions in the study group and according to sex.” Figures, currently numbered as 7 and 8, contain the values of EE for each IVGs training participant. We added a legend with a description of each participant's color. It seems to be more clear and understandable that the data charts concerns values of individual people participating in the IVGs intervention.
- We standardized the appearance of Tables 6 and 7 so that the information contained in them was clear and easy to compare.
- We completed the "Discussion" section with more articles (more than 10), including those that are not older than 10 years. We reorganized this section and we also made paragraphs. I apologize for the earlier lack of paragraphs and difficulties in reading - this is due to an error, when we were uploading the text into the journal format. In the "Discussion", we emphasized the main idea of the article, as well as its innovative character.
- We changed the second paragraph of the conclusions: “In early stage of cancer treatment, children from the study group undertook physical activity during IVGs interventions and fulfilled the MVPA recommendations. The results of follow-up study, conducted 14 months after the IVGs program showed that children continue regular activity, and their PA level was even comparable than during training intervention (no statistically significant difference in PA between post-intervention and in follow-up study). After 14 months of the IVGs intervention children were not cured as intensively as during first stage of cancer treatment and their fitness parameters were better, as shown by the CPET (the test duration was pro-longed and the peak heart rate was higher). As a result, cancer treated children had the opportunity to be much more active during this relative recovery period. This may indicate the necessity of sustained rehabilitation programs for whole time of childhood cancer treatment.
- We added a few more recent references from the last few years (<10 years). We cited them in the "Introduction" section to describe the available methods with the use of interactive technologies in modern rehabilitation. We also supplemented the "Discussion" with more literature from recent years.
- Additional supplementary files: It is possible that there were some supplementary files that were sent by the Assistant of the Editor (consent of the legal guardian or parent and information about the bioethics committee). The Assistant of the Editor asked me to send it and I think they sent it for review as well.
Additional problems:
- We developed and explained the abbreviations of protocols in the treatment of leukemia in subsection “2.3 Participants characteristics”: “The subjects treated for ALL were included in the International collaborative treatment protocol for children and adolescents with acute lymphoblastic leukemia (AIEOP-BFM ALL 2017 protocol) [27] (n=13), whereas those treated for AML were included in the International therapeutic protocol for children with acute myeloid leukemia (AML-BFM 2012 protocol) [28] (n= 8).”
- We standardized the tables and improved their appearance.
- We corrected and developed the abbreviations used throughout the text.
- We corrected the names of the sections and subsections regarding the material and the research methods.
- We added the references needed to explain CPET: “Among the methods for the assessment of exercise tolerance which show the level of cardiorespiratory fitness, the most reliable and commonly used is the measurement of peak oxygen uptake (VO2peak) by means of respiratory gas analysis performed during the gradually increasing load known as the Cardio Pulmonary Exercise Test (CPET) [29].”
- We change the description in subsection "IVGs training intervention". We explained that the training parameters were based on baseline CPET results: "The amount of exercise load during IVGs training was determined by the results of the baseline CPET."
- Below the tables, we put a description in the form of “Note”, not “ * ”.
- The results in the current Figure 3. refer to the time before the intervention while those in Figure 9. refer to the results after the IVGs intervention. We corrected the error in the description of the figures.
- We included the HBSC questionnaire questions from Table 8. in the description of the research methods. We simplified the table and now it is easier to read. We also improved the table description: statistical significance not correlation.
Round 2
Reviewer 1 Report
Thank you for your explanations
Author Response
Thank you very much for re-evaluating our article and accepting all the changes to make our article better.
Reviewer 3 Report
First of all, I would like to congratulate the improvement of the text. I still have some small issues, which I have to mentioned. Some of them probably should be discussed with Editors, regarding the editorial standards using by Journal, another I notice reading again (but a few times, to correct everything carefully).
I thing that it is better to name all group of children as "study group" and divided it on eg. "intervention group" and "control group" in other way the readers can be confused - I suggest to corrected it carefully in all text and sorry that I haven't notice it before...
I've ask about the information of EE, and the figures are prepared better, but in text it is written that there are differences between boys and girls and test1 and test2 of EE - I could not find any statistics on it. Mayby it is wort to put it (EE) in table 6 and 7?
Text should be carefully checked for some editorial correction like eg.:
p.7 (? the number on all is 7, in my version p.2) line 48 - useless space after [14], or
line 50 strikethrough; or p.3 line 93, line114 strikethrough and others further in the text, or
line 101, 490, 658, 671 too much space between paragraphs.
Figure 4 - line with III* should be upper the dots like on the Fig.5
Tables: they are still not the best version which they should be (eg. why are tables 3 and 5 still in different manner? They show the same information before and after intervention...); see first which I've checked from the Journal website: Home-Based Intermittent Pneumatic Compression Therapy: The Impact in Chronic Leg Lymphedema in Patients Treated for Gynecologic Cancer - they are really good prepared tabele there, comparable to yours.
In text which you uploaded there are still ald and new figures and tables - I cannot read all text, they are on it - it should be corrected.
I am really glad that I was able to participate in the review of an interesting and important topic for health care.
Author Response
Thank you very much for re-evaluating our article. We took into account all the proposed changes.
- We changed the name of the groups. All the children qualified for the study we named by the "study group". We named the earlier “study group” as an "intervention group", while the “control group” remained the “control group”.
- Now, Table 4 also includes additional information about the statistical significance of the differences in the results between the group of boys and the group of girls from the intervention group. This information was not included in the text before but we mentioned it in the “Results” section. We think that this information should be in this table (Table 4.) because the reader could easily use it and have quick access to it.
- We made editorial corrections throughout the text. We apologize for these errors. Many of these were due to the "Track Changes" feature that was required by the publisher. We removed unnecessary spaces, strikethroughs and offsets.
- We corrected the tables. Currently, the tables are uniform and have the same line thickness.
- There are no previous figures in the current version of the article. The presence of the previous figures was again due to the required "Track Changes" function. Now the text is clear.
Thank you once again for analyzing our text in such a detailed manner. All the comments were very helpful for us and we are grateful to have such a Reviewer. Thanks to all the corrections made in the text, we also believe that the article became more valuable.